# LEARNING TO SOLVE CIRCUIT-SAT: AN UNSUPERVISED DIFFERENTIABLE APPROACH

**Saeed Amizadeh, Sergiy Matusevych, Markus Weimer**
Microsoft
Redmond, WA 98052
{saamizad,sergiym,Markus.Weimer}@microsoft.com

## ABSTRACT

Recent efforts to combine Representation Learning with Formal Methods, commonly known as Neuro-Symbolic Methods, have given rise to a new trend of applying rich neural architectures to solve classical combinatorial optimization problems. In this paper, we propose a neural framework that can learn to solve the Circuit Satisfiability problem. Our framework is built upon two fundamental contributions: a rich embedding architecture that encodes the problem structure, and an end-to-end differentiable training procedure that mimics Reinforcement Learning and trains the model *directly* toward solving the SAT problem. The experimental results show the superior out-of-sample generalization performance of our framework compared to the recently developed NeuroSAT method.

## 1 INTRODUCTION

Recent advances in neural network models for discrete structures have given rise to a new field in Representation Learning known as the *Neuro-Symbolic* methods. Generally speaking, these methods aim at marrying the classical symbolic techniques in Formal Methods and Computer Science to Deep Learning in order to benefit both disciplines. One of the most exciting outcomes of this marriage is the emergence of neural models for learning how to solve the classical combinatorial optimization problems in Computer Science. The key observation behind many of these models is that in practice, for a given class of combinatorial problems in a specific domain, the problem instances are typically drawn from a certain (unknown) distribution. Therefore if a sufficient number of problem instances are available, then in principle, Statistical Learning should be able to extract the common structures among these instances and produce meta-algorithms (or models) that would, in theory, outperform the carefully hand-crafted algorithms.

There have been two main approaches to realize this idea in practice. In the first group of methods, the general template of the solver algorithm (which is typically the greedy strategy) is directly imported from the classical heuristic search algorithm, and the Deep Learning component is only tasked to learn the optimal heuristics *within* this template. In combination with Reinforcement Learning, such strategy has been shown to be quite effective for various NP-complete problems – e.g. Khalil et al. (2017). Nevertheless, the resulted model is bounded by the greedy strategy, which is sub-optimal in general. The alternative is to go one step further and let Deep Learning figure out the entire solution structure from scratch. This approach is quite attractive as it allows the model not only learn the optimal (implicit) decision heuristics but also the optimal search strategies beyond the greedy strategy. However, this comes at a price: training such models can be quite challenging! To do so, a typical candidate is Reinforcement Learning (Policy Gradient, in specific), but such techniques are usually sample inefficient – e.g. Bello et al. (2016). As an alternative method for training, more recently Selsam et al. (2018) have proposed using the latent representations learned for the binary classification of the Satisfiability (SAT) problem to actually produce a neural SAT solver model. Even though using such proxy for learning a SAT solver is an interesting observation and provides us with an end-to-end differentiable architecture, the model is not directly trained toward solving a SAT problem (unlike Reinforcement Learning). As we will see later in this paper, that can indeed result in poor generalization and sub-optimal models.

In this paper, we propose a neural Circuit-SAT solver framework that effectively belongs to the second class above; that is, it learns the entire solution structure from scratch. More importantly, to train such model, we propose a training strategy that, unlike the typical Policy Gradient, is differentiable end-to-end, yet it trains the model *directly* toward the end goal (similar to Policy Gradient). Furthermore, our proposed training strategy enjoys an Explore-Exploit mechanism for better optimization even though it is not exactly a Reinforcement Learning approach.

The other aspect of building neural models for solving combinatorial optimization problems is how the problem instance should be represented by the model. Using classical architectures like RNNs or LSTMs completely ignores the inherent structure present in the problem instances. For this very reason, there has been recently a strong push to employ structure-aware architectures such as different variations of neural graph embedding. Most neural graph embedding methodologies are based on the idea of synchronously *propagating* local information on an underlying (undirected) graph that represents the problem structure. The intuition behind using local information propagation for embedding comes from the fact that many original combinatorial optimization algorithms can actually be seen propagating information. In our case, since we are dealing with Boolean circuits and circuit are Directed Acyclic Graphs (DAG), we would need an embedding architecture that take into account the special architecture of DAGs (i.e. the topological order of the nodes). In particular, we note that in many DAG-structured problems (such as circuits, computational graphs, query DAGs, etc.), the information is propagated *sequentially* rather than synchronously, hence a justification to have sequential propagation for the embedding as well. To this end, we propose a rich embedding architecture that implements such propagation mechanism for DAGs. As we see in this paper, our proposed architecture is capable of harnessing the structural information in the input circuits. To summarize, our contributions in this work are three-fold:

(a) We propose a general, rich graph embedding architecture that implements sequential propagation for DAG-structured data.

(b) We adapt our proposed architecture to design a neural Circuit-SAT solver which is capable of harnessing structural signals in the input circuits to learn a SAT solver.

(c) We propose a training strategy for our architecture that is end-to-end differentiable, yet similar to Reinforcement Learning techniques, it directly trains our model toward solving the SAT problem with an Explore-Exploit mechanism.

The experimental results show the superior performance of our framework especially in terms of generalizing to new problem domains compared to the baseline.

## 2 RELATED WORK

Deep learning on graph-structured data has recently become a hot topic in the Machine Learning community under the general umbrella of *Geometric Deep Learning* Bronstein et al. (2017). Based on the assumptions they make, these models typically divide into two main categories. In the first category, the graph-structured datapoints are assumed to share the same underlying graph structure (aka the *domain*) and only differ based on the feature values assigned to each node or edge. The methods in this category operate in both the spatial and the frequency domains; for example, Spectral CNN Bruna et al. (2013), Graph CNN Defferrard et al. (2016), Graph Neural Network Scarselli et al. (2009) and Covariant Compositional Networks Kondor et al. (2018). In the second category on the other hand, each example in the training data has its own domain (graph structure). Since the domain is varying across datapoints, these other methods mostly operate in the spatial domain and typically can be seen as the generalization of the classical CNNs (*e.g.*MoNet Monti et al. (2017)) or the classical RNNs (*e.g.*TreeLSTM Tai et al. (2015), DAG-RNN Baldi & Pollastri (2003); Shuai et al. (2016)) or both (*e.g.*GGS-NN Li et al. (2015)) to the graph domain. In this paper, we extend the single layer DAG-RNN model for DAG-structured data Baldi & Pollastri (2003); Shuai et al. (2016) to the more general deep version with Gated Recurrent Units, where each layer processes the input DAG either in the forward or the backward direction.

On the other hand, the application of Machine Learning (deep learning in specific) to logic and symbolic computation has recently emerged as a bridge between Machine Learning and the classical Computer Science. While works such as Evans et al. (2018); Arabshahi et al. (2018) have shown the effectiveness of (recursive) neural networks in modeling symbolic expressions, others have

taken one step further and tried to learn approximate algorithms to solve symbolic NP-complete problems Khalil et al. (2017); Bello et al. (2016); Vinyals et al. (2015). In particular, as opposed to black box methods (*e.g.* Bello et al. (2016); Vinyals et al. (2015)), Khalil *et al.* Khalil et al. (2017) have shown that by incorporating the underlying graph structure of a NP-hard problem, efficient search heuristics can be learned for the greedy search algorithm. Although working in the context of greedy search introduces an inductive bias that benefits the sample efficiency of the framework, the resulted algorithm is still bounded by the sub-optimality of the greedy search. More recently, Selsal *et al.* Selsam et al. (2018) have introduced the *NeuroSAT* framework - a deep learning model aiming at learning to solve the Boolean Satisfiability problem (SAT) from scratch without biasing it toward the greedy search. In particular, they have primarily approached the SAT problem as a binary classification problem and proposed a clustering-based post-processing analysis to find a SAT solution from the latent representations extracted from the learned classifier. Although, they have shown the empirical merits of their proposed framework, it is not clear why the proposed post-processing clusetring should find the SAT solution without being explicitly trained toward that goal. In this paper, we propose a deep learning framework for the Circuit-SAT problem (a more general form of the SAT problem), but in contrast to NeuroSAT, our model is *directly* trained toward finding SAT solutions *without requiring to see them in the training sample.*

## 3   DAG EMBEDDING

In this section, we formally formulate the problem of learning on DAG-structured data and propose a deep learning framework to approach the problem. It should be noted that even though this framework has been developed for DAGs, the underlying dataset can be a general graph as long as an explicit ordering for the nodes of each graph is available. This ordering is naturally induced by the topological sort algorithm in DAGs or can be imposed on general undirected graphs to yield DAGs.

### 3.1   NOTATIONS AND DEFINITIONS

Let $G = \langle V_G, E_G \rangle$ denote a Directed Acyclic Graph (DAG). We assume the the set of nodes of $G$ are ordered according to the topological sort of the DAG. For any node $v \in V_G$, $\pi_G(v)$ represents the set of direct predecessors of $v$ in $G$. Also for a given DAG $G$, we define the *reversed* DAG, $G^r$ with the same set of nodes but reversed edges. When topologically sorted, the nodes of $G^r$ appear in the reversed order of those of $G$. Furthermore, for a given $G$, let $\mu_G : V_G \mapsto \mathbb{R}^d$ be a $d$-dimensional vector function defined on the nodes of $G$. We refer to $\mu_G$ as a *DAG function* – i.e. a function that is defined on a DAG. Note that the notation $\mu_G$ implicitly induces the DAG structure $G$ along with the vector function defined on the DAG. Figure 1(a) shows an example DAG function with $d = 3$. Finally, let $\mathcal{G}^d$ denote the space of all possible $d$-dimensional functions $\mu_G$ (along with their underlying graphs $G$). We define the parametric functional $\mathcal{F}_{\boldsymbol{\theta}} : \mathcal{G}^d \mapsto \mathcal{O}$ that maps any function $\mu_G$ (defined on some DAG $G$) in $\mathcal{G}^d$ to some output space $\mathcal{O}$.

### 3.2   THE GENERAL MODEL

The next step is to define the mathematical form of the functional $\mathcal{F}_{\boldsymbol{\theta}}$. In this work, we propose:

$$\mathcal{F}_{\boldsymbol{\theta}}(\mu_G) = \mathcal{C}_{\boldsymbol{\alpha}}\bigg( \mathcal{P}\big(\mathcal{E}_{\boldsymbol{\beta}}(\mu_G)\big) \bigg) \tag{1}$$

Intuitively, $\mathcal{E}_{\boldsymbol{\beta}} : \mathcal{G}^d \mapsto \mathcal{G}^q$ is the *embedding* function that maps the input $d$-dimensional DAG functions into a $q$-dimensional DAG function space. Note that the embedding function in general may transform both the underlying DAG size/structure as well as the the DAG function defined on it. In this paper, however, we assume it only transforms the DAG function and keeps the input DAG structure intact. Once the DAG is embedded into the new space, we apply the fixed *pooling* function $\mathcal{P} : \mathcal{G}^q \mapsto \mathcal{G}^q$ on the embedded DAG function to produce a (possibly) aggregated version of it. For example, if we are interested in DAG-level predictions, $\mathcal{P}$ can be average pooling across all nodes of the input DAG to produce a singleton DAG; whereas, in the case of node-level predictions, $\mathcal{P}$ is simply the Identity function. In this paper, we set $\mathcal{P}$ to retrieve only the *sink* nodes in the input DAG. Finally, the *classification* function $\mathcal{C}_{\boldsymbol{\alpha}} : \mathcal{G}^q \mapsto \mathcal{O}$ is applied on the aggregated DAG function to produce the final prediction output in $\mathcal{O}$. In this work, we set $\mathcal{C}_{\boldsymbol{\alpha}}$ to be a multi-layer neural network. The tuple $\boldsymbol{\theta} = \langle \boldsymbol{\alpha}, \boldsymbol{\beta} \rangle$ identifies all the free parameters of the model.

### 3.3 THE DAG EMBEDDING LAYER

The (supervised) embedding of graph-based data into the traditional vector spaces has been a hot topic recently in the Machine Learning community Li et al. (2015); Shuai et al. (2016); Tai et al. (2015). Many of these frameworks are based on the key idea of representing each node in the input graph by a latent vector called the node *state* and update these latent states via an iterative (synchronous) propagation mechanism that takes the graph structure into account. Two of these methodologies that are closely related to the proposed framework in this paper are the Gated Graph Sequence Neural Networks (GGS-NN) Li et al. (2015) and DAG Recurrent Neural Networks (DAG-RNN) Shuai et al. (2016). While GGS-NNs apply multi-level *Gated Recurrent Unit (GRU)* like updates in an iterative propagation scheme on general (undirected) graphs, DAG-RNNs apply simple RNN logic in a one-pass, sequential propagation mechanism from the input DAG's source nodes to its sink nodes.

Our proposed framework is built upon the DAG-RNN framework Shuai et al. (2016) but it enriches this framework further by incorporating key ideas from GGS-NNs Li et al. (2015), Deep RNNs Pascanu et al. (2013) and sequence-to-sequence learning Sutskever et al. (2014). Before we explain our framework, it is worth noting that assiging input feature/state vectors to each node is equivalent to defining a DAG function in our framework. For the sake of notational simplicity, for the input DAG function $\mu_G$, we define the $d$-dimensional node feature vector $\boldsymbol{x}_v = \mu_G(v)$ and the $q$-dimensional node state vector $\boldsymbol{h}_v = \delta_G(v)$ for some unknown DAG function $\delta_G : V_G \mapsto \mathbb{R}^q$. Given the node feature vectors $\boldsymbol{x}_v$ for an input DAG, the update rule for the state vector at each node is defined as:

$$\boldsymbol{h}_v = GRU(\boldsymbol{x}_v, \boldsymbol{h}'_v), \text{ where } \boldsymbol{h}'_v = \mathcal{A}\big(\{\boldsymbol{h}_u \mid u \in \pi(v)\}\big) \qquad (2)$$

where $GRU(.)$ is the standard GRU Chung et al. (2014) function applied on the input vector at node $v$ and the *aggregated* state of its direct predecessors which in turn is computed by the *aggregator* function $\mathcal{A} : 2^{V_G} \mapsto \mathbb{R}^q$. The aggregator function is defined as a tunable *deep set* function Zaheer et al. (2017) with free parameters that is invariant to the permutation of its inputs. The main difference between these proposed updates rules and the ones in DAG-RNN is in DAG-RNN, we have the simple RNN logic instead of GRU, and the aggregation logic is simply (fixed) summation.

By applying the update logic in equation 2 sequentially on the nodes of the input DAG processed in the topological sort order, we compute the state vector $\boldsymbol{h}_v$ for all nodes of $G$ in one pass. This would complete the one layer (forward) embedding of the input DAG function, or $\mathcal{E}_{\boldsymbol{\beta}}(\mu_G) = \delta_G$. Note that the same way that DAG-RNNs are the generalization of RNNs on sequences to DAGs, our proposed one-layer embedding can be seen as the generalization of GRU-NNs on sequences to DAGs.

Furthermore, we introduce the *reversed* layers (denoted by $\mathcal{E}^r$) that are similar to the regular forward layers except that the input DAG is processed in the reversed order. Alternatively, reversed layers can be seen as regular layers that process the reversed version of the input DAG $G^r$; that is, $\mathcal{E}^r(\mu_G) \equiv \mathcal{E}(\mu_{G^r})$. The main reason we have introduced reversed layers in our framework is because in the regular forward layers, the state vector for each node is only affected by the information flowing from its ancestor nodes; whereas, the information from the descendant nodes can also be highly useful for the learning task in hand. The reversed layers provide such information for the learning task. Furthermore, the introduction of reversed layers is partly motivated by the successful application of processing sequences backwards in sequence-to-sequence learning Sutskever et al. (2014). Sequences can be seen as special-case linear DAGs; as a result, reversed layers can be interpreted as the generalized version of reversing sequences.

### 3.4 DEEP-GATED DAG RECURSIVE NEURAL NETWORKS

The natural extension of the one-layer embedding is the stacked $L$-layer version where the $i$th layer has its own parameters $\boldsymbol{\beta_i}$ and output DAG function dimensionality $q_i$. Furthermore, the stacked $L$ layers can be sequentially applied $T$ times in the recurrent fashion to generate the final embedding:

$$\mathcal{E}_{\boldsymbol{\beta}}(\mu_G) \equiv \mathcal{E}_{\boldsymbol{\beta}}^T(\mu_G), \text{ where } \mathcal{E}_{\boldsymbol{\beta}}^t(\mu_G) = \mathcal{E}_{stack}\big(Proj_{\boldsymbol{H}}(\mathcal{E}_{\boldsymbol{\beta}}^{t-1}(\mu_G))\big), \forall t \in 2..T \qquad (3)$$

$$\mathcal{E}_{\boldsymbol{\beta}}^1(\mu_G) = \mathcal{E}_{stack}(\mu_G) \qquad (4)$$

$$\text{s.t. } \mathcal{E}_{stack} = \mathcal{E}_{\boldsymbol{\beta_L}} \circ \mathcal{E}_{\boldsymbol{\beta_{L-1}}} \circ ... \circ \mathcal{E}_{\boldsymbol{\beta_1}} \qquad (5)$$

where $\boldsymbol{\beta} = \langle \boldsymbol{\beta_1}, ..., \boldsymbol{\beta_L}, \boldsymbol{H} \rangle$ is the list of the parameters and $Proj_{\boldsymbol{H}} : \mathcal{G}^{q_L} \mapsto \mathcal{G}^d$ is a linear projection with the projection matrix $\boldsymbol{H}_{d \times q_L}$ that simply adjusts the output dimensionality of $\mathcal{E}_{stack}$ so it can

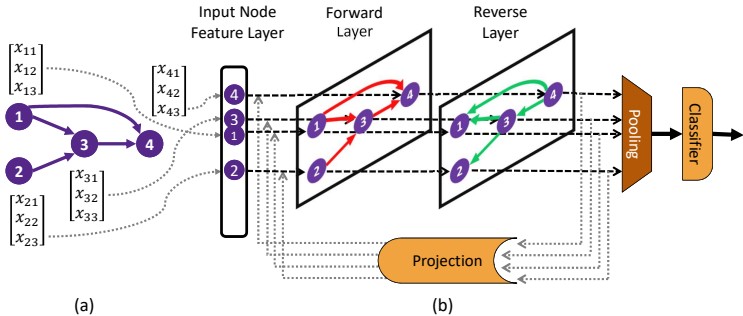

Figure 1: (a) A toy example input DAG function $\mu_G$, (b) a DG-DAGRNN model that processes the input in (a) using two sequential DAG embedding layers: a forward layer followed by a reverse layer. The solid red and green arrows show the flow of information within each layer while the black arrows show the feed-forward flow of information in between the layers. Also, the dotted blue arrows show the recurrent flow of information from the last embedding layer back to the first one.

be fed back to $\mathcal{E}_{stack}$ as the input. In our experiments, we have found that by letting $T > 1$, we can significantly improve the accuracy of our models *without* introducing more trainable parameters. In practice, we fix the value of $T$ during training and increase it during testing to achieve better accuracy. Also note that the $L$ stacked layers in $\mathcal{E}_{stack}$ can be any permutation of regular and reversed layers. We refer to this proposed framework as *Deep-Gated DAG Recursive Neural Networks* or DG-DAGRNN for short. Figure 1(b) shows an example 2-layer DG-DAGRNN model with one forward layer followed by a reversed layer.

## 4 APPLICATION TO THE CIRCUIT-SAT PROBLEM

The Circuit Satisfiability problem (aka Circuit-SAT) is a fundamental NP-complete problem in Computer Science. The problem is defined as follows: given a Boolean expression consists of Boolean variables, parentheses, and logical gates (specifically **And** $\wedge$, **Or** $\vee$ and **Not** $\neg$), find an assignment to the variables such that it would satisfy the original expression, aka a solution. If the expression is not satisfiable, it will be labeled as UNSAT. Moreover, when represented in the circuit format, Boolean expressions can aggregate the repetitions of the same Boolean sub-expression in the expression into one node in the circuit. This is also crucial from the learning perspective as we do not want to learn two different representations for the same Boolean sub-expression.

In this section, we apply the framework from the previous section to learn a Circuit-SAT solver merely from data. More formally, a Boolean circuit can be modeled as a DAG function $\mu_G$ with each node representing either a Boolean variable or a logical gate. In particular, we have $\mu_G : V_G \mapsto \mathbb{R}^4$ defined as $\mu_G(v) = \text{One-Hot}(type(v))$, where $type(v) \in \{\textbf{And}, \textbf{Or}, \textbf{Not}, \textbf{Variable}\}$. All the source nodes in a circuit $\mu_G$ have type **Variable**. Moreover, each circuit DAG has only one sink node (the root node of the Boolean expression).

Similar to Selsam et al. (2018), we could also approach the Circuit-SAT problem from two different angles: (1) predicting the circuit satisfiability problem as a binary classification problem, and (2) solving the Circuit-SAT problem directly by generating a solution if the input circuit is indeed SAT. In Selsam et al. (2018), solving the former is the prerequisite for solving the latter. However, that is not the case in our proposed model and since we are interested to actually solve the SAT problems, we do not focus on the binary classification problem. Nevertheless, our model can be easily adapted for SAT classification, as illustrated in Appendix A.

### 4.1 NEURAL CIRCUIT-SAT SOLVER

Learning to solve SAT problems (*i.e.* finding a satisfying assignment) is indeed a much harder problem than SAT/UNSAT classification. In the NeuroSAT framework, Selsam et al. (2018), the authors have proposed a post-processing unsupervised procedure to decode a solution from the latent state representations of the Boolean literals. Although this approach works empirically for many SAT

problems, it is not clear that it would also work for the Circuit-SAT problems. But more importantly, it is not clear why this approach should decode the SAT problems in the first place because the objective function used in Selsam et al. (2018) does not explicitly contain any component for solving SAT problems; in fact, the decoding procedure is added as a secondary analysis *after* training. In other words, the model is not optimized toward actually finding SAT assignments.

In contrast, in this paper, we pursue a completely different strategy for training a neural Circuit-SAT solver. In particular, using the DG-DAGRNN framework, we learn a neural functional $\mathcal{F}_{\boldsymbol{\theta}}$ on the space of circuits $\mu_G$ such that given an input circuit, it would directly generate a satisfying assignment for the circuit if it is indeed SAT. Moreover, we explicitly train $\mathcal{F}_{\boldsymbol{\theta}}$ to generate SAT solutions *without requiring to see any actual SAT assignment during training.* Our proposed strategy for training $\mathcal{F}_{\boldsymbol{\theta}}$ is reminiscent of Policy Gradient methods in Deep Reinforcement Learning Arulkumaran et al. (2017).

**The Solver Network.** We start with characterizing the components of $\mathcal{F}_{\boldsymbol{\theta}}$. First, the embedding function $\mathcal{E}_{\boldsymbol{\beta}}$ is set to be a multi-layer recursive embedding as in equation 3 with interleaving regular forward and reversed layers making sure that the last layer is a reversed layer so that we can read off the final outputs of the embedding from the **Variable** nodes (i.e. the sink nodes of the reversed DAG). The classification function $\mathcal{C}_{\boldsymbol{\alpha}}$ is set to be a MLP with ReLU activation function for the hidden layers and the Sigmoid activation for the output layer. The output space here encodes the *soft* assignment (i.e. in range $[0, 1]$) to the corresponding variable node in the input circuit. We also refer to $\mathcal{F}_{\boldsymbol{\theta}}$ as the *solver* or the *policy* network.

**The Evaluator Network.** Furthermore, for any given circuit $\mu_G$, we define the *soft evaluation* function $\mathcal{R}_G$ as a DAG computational graph that shares the same topology $G$ with the circuit $\mu_G$ except that the **And** nodes are replaced by the *smooth min* function, the **Or** nodes by the *smooth max* function and the **Not** nodes by $\mathcal{N}(z) = 1 - z$ function, where $z$ is the input to the **Not** node. The smooth min and max functions are defined as:

$$S_{max}(a_1, a_2, ..., a_n) = \frac{\sum_{i=1}^{n} a_i e^{a_i/\tau}}{\sum_{i=1}^{n} e^{a_i/\tau}}, \; S_{min}(a_1, a_2, ..., a_n) = \frac{\sum_{i=1}^{n} a_i e^{-a_i/\tau}}{\sum_{i=1}^{n} e^{-a_i/\tau}} \tag{6}$$

where $\tau \geq 0$ is the *temperature*. For $\tau = +\infty$, both $S_{max}()$ and $S_{min}()$ are the arithmetic mean functions. As $\tau \to 0$, we have $S_{max}() \to \max()$ and $S_{min}() \to \min()$. One can also show that $\forall \boldsymbol{a} = (a_1, ..., a_n) : \min(\boldsymbol{a}) < S_{min}(\boldsymbol{a}) < S_{max}(\boldsymbol{a}) < \max(\boldsymbol{a})$. More importantly, as opposed to the $\min()$ and $\max()$ functions, their smooth versions are fully differentiable w.r.t. *all* of their inputs.

As its name suggests, the soft evaluation function evaluates a soft assignment (i.e. in $[0, 1]$) to the variables of the circuit. In particular, at a low enough temperature, if for a given input assignment, $\mathcal{R}_G$ yields a value *strictly* greater than $0.5$, then that assignment (or its hard counterpart) can be seen as a satisfying solution for the circuit. We also refer to $\mathcal{R}_G$ as the *evaluator* or the *reward* network. Note that the evaluator network does *not* have any trainable parameter.

Encoding logical expressions into neural networks is not new per se as there has been recently a push to enrich deep learning with symbolic computing Hu et al. (2016); Xu et al. (2017). What are new in our framework, however, are two folds: (a) each graph example in our dataset induces a different evaluation network as opposed to having one fixed network for the entire dataset, and (b) by encoding the logical operators as smooth min and max functions, we provide a more efficient framework for back-propagating the gradients and speeding up the learning as the result, as we will see shortly.

**The Optimization.** Putting the two pieces together, we define the *satisfiability* function $\mathcal{S}_{\boldsymbol{\theta}} : \mathcal{G} \mapsto [0, 1]$ as: $\mathcal{S}_{\boldsymbol{\theta}}(\mu_G) = \mathcal{R}_G\big(\mathcal{F}_{\boldsymbol{\theta}}(\mu_G)\big)$. Intuitively, the satisfiability function uses the solver network to produce an assignment for the input circuit and then feeds the resulted assignment to the evaluator network to see if it satisfies the circuit. We refer to the final output of $\mathcal{S}_{\boldsymbol{\theta}}$ as the *satisfiability value* for the input circuit, which is a real number in $[0, 1]$. Having computed the satisfiability value, we define the loss function as the smooth Step function:

$$\mathcal{L}(s) = \frac{(1 - s)^{\kappa}}{(1 - s)^{\kappa} + s^{\kappa}} \tag{7}$$

where $s = \mathcal{S}_{\boldsymbol{\theta}}(\mu_G)$ and $\kappa \geq 1$ is a constant. By minimizing the loss function in equation 7, we push the solver network to produce an assignment that yields a higher satisfiability value $\mathcal{S}_{\boldsymbol{\theta}}(\mu_G)$. For satisfiable circuits this would eventually result in finding a satisfiable assignment for the circuit. However, if the input circuit is UNSAT, the maximum achievable value for $\mathcal{S}_{\boldsymbol{\theta}}$ is $0.5$ as we have

shown in Appendix B. In practice though, the inclusion of UNSAT circuits in the training data slows down the training process mainly because the UNSAT circuits keep confusing the solver network as it tries hard to find a SAT solution for them. For this very reason, in this scheme, we only train on SAT examples and exclude the UNSAT circuits from training. Nevertheless, if the model has enough capacity, one can still include the UNSAT examples and pursue the training as a pure unsupervised learning task since the true SAT/UNSAT labels are not used anywhere in equation 7.

Moreover, the loss function in equation 7 has a nice property of having higher gradients for satisfiability values close to 0.5 when $\kappa > 1$ (we set $\kappa = 10$ in our experiments). This means that the gradient vector in backpropagation is always dominated by the examples closer to the decision boundary. In practice, that would mean that the training algorithm immediately in the beginning pushes the easier examples in the training set (with satisfiability values close to 0.5) to the SAT region ($> 0.5$) with a safety margin from 0.5. As the training progresses, harder examples (with satisfiability values close to 0) start moving toward the SAT region.

As mentioned before, the proposed learning scheme in this section can be seen as a variant of Policy Gradient methods, where the solver network represents the policy function and the evaluator network acts as the reward function. The main difference here is that in our problem the mathematical form of the reward function is fully known and is differentiable so the entire pipeline can be trained using backpropagation in an end-to-end fashion to maximize the total reward over the training sample.

**Exploration vs. Exploitation.** The reason we use the smooth min and max functions in the evaluator network instead of the actual $\min()$ and $\max()$ is that in a min-max circuit, the gradient vector of the output of the circuit w.r.t. its inputs has at most one non-zero entry [1]. That is, the circuit output is sensitive to only one of its inputs in the case of infinitesimal changes. For fixed input values, we refer to this input as the *active input* and to the path from the active input to the output as the *active path*. In the case of a min-max evaluator, the gradients flow back only through the active path of the evaluator network forcing the solver network to change such that it can satisfy the input circuit through its active path only. This strategy however is quite myopic and, as we observed empirically, leads to slow training and sub-optimal solutions. To avoid this effect, we use the smooth min and max functions in the evaluator network to allow the gradients to flow through *all* paths in the input circuit. Furthermore, in the beginning of the training we start with a high temperature value to let the model *explore* all paths in the input circuits for finding a SAT solution. As the training progresses, we slowly anneal the temperature toward 0 so that the model *exploits* more active path(s) for finding a solution. One annealing strategy is to let $\tau = t^{-\epsilon}$, where $t$ is timestep and $\epsilon$ is the annealing rate. In our experiments we set $\epsilon = 0.4$. It should be noted that at the test time, the smooth min and max functions are replaced by their non-smooth versions.

**Prediction.** Given a test circuit $\mu_G$, we evaluate $s = \mathcal{S}_{\boldsymbol{\theta}}(\mu_G) = \mathcal{R}_G\big(\mathcal{F}_{\boldsymbol{\theta}}(\mu_G)\big)$. If $s > 0.5$ then the circuit is classified as SAT and the SAT solution is provided by $\mathcal{F}_{\boldsymbol{\theta}}(\mu_G)$. Otherwise, the circuit is classified as UNSAT. This way, unlike SAT classification, we predict SAT for a given circuit only if we have already found a SAT solution for it. In other words, *our model never produces false positives*. We have formally proved this in Appendix B. Moreover, at the prediction time, we do not need to set the number of recurrences $T$ in equation 3 to the same value we used for training. In fact, we have observed by letting $T$ to be variable on the per example basis, we can improve the model accuracy quite significantly at the test time.

## 5 EXPERIMENTAL EVALUATION

The baseline method we have compared our framework to is the NeuroSAT model by Selsam et al. (2018). Like most classical SAT solvers, NeuroSAT assumes the input problem is given in the *Conjunctive Normal Form* (CNF). Even though that is a fair assumption in general, in some cases, the input does not naturally come as CNF. For instance, in hardware verification, the input problems are often in the form of circuits. One can indeed convert the circuit format into CNF in polynomial time using Tseitin transformation. However, such transformation will introduce extra variables (i.e. the derived variables) which may further complicate the problem for the SAT solver. More importantly, as a number of works in the SAT community have shown, such transformations typically lose the structural information embedded in the circuit format, which otherwise can be a rich source of

---

[1] Assuming there are no ties among the circuit inputs.

information for the SAT solver, Thiffault et al. (2004); Andrews (2002); Biere (2008); Fu & Malik (2007); Velev (2007). As a result, there has been quite an effort in the classical SAT community to develop SAT solvers that directly work with the circuit format Thiffault et al. (2004); Jain & Clarke (2009). In the similar vein, our neural framework for learning a SAT solver enables us to harness such structural signals in learning by directly consuming the circuit format. That contrasts the NeuroSAT approach which cannot in principle benefit from such structural information.

Despite this clear advantage of our framework to NeuroSAT, in this paper, we assume the (raw) input problems come in CNF, just so we can make a fair comparison to NeuroSAT. Instead for our method, we propose to use pre-processing methods to convert the input CNF into circuit that has the potential of injecting structural information into the circuit structure. In particular, if available, one can in principle encode problem-specific heuristics into the structure while building the circuit. For example, if there is a variable ordering heuristic available for a specific class of SAT problems, it can be used to build that target circuit in a certain way, as discussed in Appendix C. Note that we could just consume the original CNF; after all, CNF is a (flat) circuit, too. But as we empirically observed, that would negatively affect the results, which again highlights the fact that our proposed framework has been optimized to utilize circuit structure as much as possible.

Both our method and NeuroSAT require a large training sample size for moderate size problems. The good news is both methods can effectively be trained on an infinite stream of randomly generated problems in real-world applications. However, since we ran our experiments only on one GPU with limited memory, we had to limit the training sample size for the purpose of experimentation. This in turn restricts the maximum problem sizes we could train both models on. Nevertheless, our method can generalize pretty well to out-of-sample SAT problems with much larger sizes, as shown below.

## 5.1 RANDOM $k$-SAT

We have used the generation process proposed in the NeuroSAT paper Selsam et al. (2018) to generate random $k$-SAT CNF pairs (with $k$ stochastically set according to the default settings in Selsam et al. (2018)). These pairs are then directly fed to NeuroSAT for training. For our method, on the other hand, we first need to convert these CNFs into circuits[2]. In Appendix C, we have described the details of this conversion process.

**Experimental Setup**: We have trained a DG-DAGRNN model (i.e. our framework) and a NeuroSAT model on a dataset of 300K SAT and UNSAT pairs generated according to the scheme proposed in Selsam et al. (2018). The number of Boolean variables in the problems in this dataset ranges from 3 to 10. We have designed both models to have roughly $\sim 180K$ tunable parameters. In particular our model has two DAG embedding layers: a forward layer followed by a reversed layer, each with the embedding dimension $q = 100$. The classifier is a 2-layer MLP with hidden dimensionality 30. The aggregator function $\mathcal{A}(\cdot)$ consists of two 2-layer MLPs, each with hidden dimensionality 50. For training, we have used the Adam optimization algorithm with learning rate of $10^{-5}$, weight decay of $10^{-10}$ and gradient clipping norm of 0.65. We have also applied a dropout mechanism for the aggregator function during training with the rate of 20%. For the NeuroSAT model, we have used the default hyper-parameter settings proposed in Selsam et al. (2018). Finally, since our model does not produce false positives, we have only included satisfiable examples in the test data for all experiments.

**In-Sample Results**: Once we trained the two models, the main performance metric we measure is the percentage of SAT problems in the test set that each model can actually find a SAT solution for.[3] Figure 2 (Left) shows this metric on a test set from the same distribution for both our model and NeuroSAT as the number of recurrences (or propagation iterations for NeuroSAT) $T$ increases. Not surprisingly, both methods are able to decode more SAT problems as we increase $T$. However, our method converges much faster than NeuroSAT (to a slightly smaller value). In other words, compared to NeuroSAT, our method requires smaller of number of iterations at the test time to decode SAT problems. We conjecture this is due to the fact the sequential propagation mechanism in DG-DAGRNN is more effective in decoding the structural information in the circuit format for the SAT problem than the synchronous propagation mechanism in NeuroSAT for the flat CNF.

---

[2]Indeed only the SAT examples because our method does not need the UNSAT examples for training, as explained in Section 4.

[3]Note that in this paper, we do *not* focus on the SAT binary classification performance.

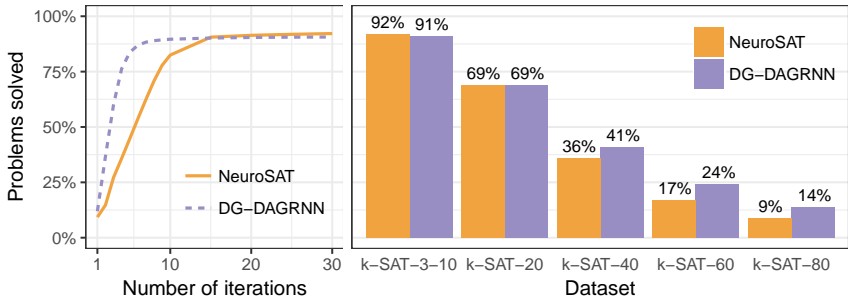

Figure 2: (Left) In-Sample test results comparing between DG-DAGRNN and NeuroSAT, as the number of recurrence iterations $T$ increases. (Right) Out-of-Sample test results comparing the two methods when tested on much larger problems.

**Out-of-Sample Results**: Furthermore, we evaluated both trained models on test datasets drawn from different distributions than the training data with much larger number of variables ($20$, $40$, $60$ and $80$ variables, in particular). We let both models iteratively run on each test dataset until the test metric converges. Figure 2 (Right) shows the test metric for both methods on these datasets after convergence. As the results demonstrate, compared to that of our method, the performance of NeuroSAT declines faster as we increase the number variables during test time, with a significant margin. In other words, our method generalizes better to out-of-sample, larger problems during the test time. We attribute this to the fact that NeuroSAT is trained toward the SAT classification problem as a proxy to learn a solver. This may result in the classifier picking up certain features that are informative for classification of in-sample examples which are, otherwise, harmful (or useless at best) for learning a solver for out-of-sample examples. Our framework, on the other hand, simply does not suffer from such problem because it is directly trained toward solving the SAT problem.

**Time Complexity**: We trained both our model and NeuroSAT for a day on a single GPU. To give an idea of the test time complexity, it took both our model and NeuroSAT roughly about 3s to run for $40$ iterations on a single example of $20$ variables. We also measured the time that it takes for a modern SAT Solver (MiniSAT here) to solve a similar example to be roughly about $0.7$s in average. Despite this difference, our neural approach is way more prallelizable compared to modern solvers such that many examples can be solved *concurrently* in a single batch on GPU. For example, while it took MiniSAT 114min to solve a set of $10,000$ examples, it took our method only 8min to solve for the same set in a batch-processing fashion on GPU. This indeed shows another important advantage of our neural approach toward SAT solving in large-scale applications: the extreme parallelization.

## 5.2    RANDOM GRAPH $k$-COLORING

To further evaluate the generalization performance of the trained models from the previous section, we have tested them on SAT problems coming from an entire different domain than $k$-SAT problems. In particular, we have chosen the graph $k$-coloring decision problem which belongs to the class of NP-complete problems. In short, given an undirected graph $G$ with $k$ color values, in graph $k$-coloring decision problem, we seek to find a mapping from the graph nodes to the color set such that no adjacent nodes in the graph have the same color. This classical problem is reducible to SAT. Moreover, the graph topology in general contains valuable information that can be further injected into the circuit structure when preparing circuit representation for our model. Appendix D illustrates how we incorporate this information to convert instances of the graph $k$-coloring problem into circuits. For this experiment, we have generated two different test datasets:

**Dataset-1**: We have generated a diverse set of random graphs with number of nodes ranging between $6$ and $10$ and the edge percentage of $37\%$. The random graphs are evenly generated according to six distinct distributions: Erdos-Renyi, Barabasi-Albert, Power Law, Random Regular, Watts-Strogatz and Newman-Watts-Strogatz. Each generated graph is then paired with a random color number in $2 \le k \le 4$ to generate a graph $k$-coloring instance. We only keep the SAT instances in the dataset.

**Dataset-2**: We first generate random trees with the same number of nodes as Dataset-1. Then each tree is paired with a random color number in $2 \le k \le 4$. Since the chromatic number of trees is 2,

every single pair so far is SAT. Lastly, for each pair we keep adding random edges to the graph until it becomes UNSAT, then we remove the last added edge to make the instance SAT again and stop.

Even though Dataset-1 has much higher coverage in terms of different graph distributions, Dataset-2 contains harder SAT examples in general, simply because in average, it contains maximally constrained instances that are still SAT. We evaluated both our method and NeuroSAT (which were both trained on $k$-SAT-3-10) on these test datasets. Our method could solve $48\%$ and $27\%$ of the SAT problems in Dataset-1 and Dataset-2, respectively. However, to our surprise, the same NeuroSAT model that generated the out-of-sample results on $k$-SAT datasets in Figure 2, could not solve any of the SAT graph $k$-coloring problems in Dataset-1 and Dataset-2, even after 128 propagation iterations. This does not match the results reported in Selsam et al. (2018) on graph coloring. We suspect different CNF formulations for the graph $k$-coloring problem might be the cause behind this discrepancy, which would mean that NeuroSAT is quite sensitive to the change of problem distribution. Nevertheless, the final judgment remains open up to further investigations.

In a separate effort, we tried to actually train a fresh NeuroSAT model on a larger versions of Dataset-1 and Dataset-2 which also included UNSAT examples. However, despite a significant decrease on the classification training loss, NeuroSAT failed to decode any of the SAT problems in the test sets. We attribute this behavior to the fact that NeuroSAT is dependent on learning a good SAT classifier that can capture the conceptual essence of SAT vs. UNSAT. As a result, in order to avoid learning superficial classification features, NeuroSAT restricts its training to a strict regime of SAT-UNSAT pairs, where the two examples in a pair only differ in negation of one literal. However, such strict regime can be only enforced in the random $k$-SAT problems. For graph coloring, the closest strategy we could come up with was the one in Dataset-2, where the SAT-UNSAT examples in a pair only differ in an edge (which still translates to a couple of clauses in the CNF). This again signifies the importance of learning the solver directly rather than relying on a classification proxy.

## 6 DISCUSSION

In this paper, we proposed a neural framework for efficiently learning a Circuit-SAT solver. Our methodology relies on two fundamental contributions: (1) a rich DAG-embedding architecture that implements the sequential propagation mechanism on DAG-structured data and is capable of learning useful representations for the input circuits, and (2) an efficient training procedure that trains the DAG-embedding architecture directly toward solving the SAT problem without requiring SAT/UNSAT labels in general. Our proposed training strategy is fully differentiable end-to-end and at the same time enjoys many features of Reinforcement Learning such as an Explore-Exploit mechanism and direct training toward the end goal.

As our experiments showed, the proposed embedding architecture is able to harness structural information in the input DAG distribution and as a result solve the test SAT cases in a fewer number of iterations compared to the baseline. This would also allow us to inject domain-specific heuristics into the circuit structure of the input data to obtain better models for that specific domain. Moreover, our direct training procedure as opposed to the indirect, classification-based method in NeuroSAT enables our model to generalize better to out-of-sample test cases, as demonstrated by the experiments. This superior generalization got even more expressed as we transferred the trained models to a complete new domain (i.e. graph coloring). Furthermore, we argued that not only does direct training give us superior out-of-sample generalization, but it is also essential for the problem domains where we cannot enforce the strict training regime where SAT and UNSAT cases come in pairs with almost identical structures, as proposed by Selsam et al. (2018).

Future efforts in this direction would include closely examining the SAT solver algorithm learned by our framework to see if any high-level knowledge and insight can be extracted to further aide the classical SAT solvers. Needless to say, this type of neural models have a long way to go in order to compete with industrial SAT solvers; nevertheless, these preliminary results are promising enough to motivate the community to pursue this direction.

### ACKNOWLEDGMENTS

We would like to thank Leonardo de Moura and Nikolaj Bjorner from Microsoft Research for the valuable feedback and discussions.

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

## APPENDIX A: ADAPTING DG-DAGRNN FOR CIRCUIT-SAT CLASSIFICATION

In the classification problem, we are interested to merely classify each input circuit as SAT or UNSAT. To do so, we customize DG-DAGRNN framework as follows. The classification function $\mathcal{C}_{\boldsymbol{\alpha}}$ is set to be a MLP with ReLU activation function for the hidden layers and the Sigmoid activation for the output layer. As the result the output space $\mathcal{O}$ will become $[0, 1]$. The embedding function $\mathcal{E}_{\boldsymbol{\beta}}$ is set to be a multi-layer recursive embedding as in equation 3 with interleaving regular forward and reversed layers. For the classification problem, we make sure the last layer of the embedding is a forward layer so that we can read off from only one sink node (i.e. the expression root node) and feed the result to the classification function for the final prediction. Finally given a labeled training set, we minimize the standard cross-entropy loss via the end-to-end backpropagation through the entire network.

## APPENDIX B: PROOF OF NO FALSE POSITIVES

In this section, we prove that for any UNSAT input circuit $\mu_G$, the satisfiability function $\mathcal{S}_{\boldsymbol{\theta}}(\mu_G)$ at the prediction time will never go beyond $0.5$, and as a result, our model would never produce false positives. To prove that, first we show that thresholding the output of the evaluator network $\mathcal{R}_G$ for a soft assignment $\boldsymbol{a}$ is equivalent to applying the original circuit $\mu_G$ to the hard assignment corresponding to $\boldsymbol{a}$:

**Lemma 1.** *Let $\mu_G$ be any DAG function representing a Boolean circuit with underlying topology $G$. Also let $\mathcal{R}_G$ be the evaluator network corresponding to $\mu_G$ where all the **And**, **Or** and **Not** gates are replaced by the $\min(\cdot)$, $\max(\cdot)$ and $\mathcal{N}(\cdot)$ functions, respectively. Moreover, for any soft assignment $\boldsymbol{a} = (a_1, a_2, ..., a_n) \in [0, 1]^n$, let its corresponding hard assignment $\mathcal{H}(\boldsymbol{a}) = \big(\mathcal{H}(a_1), \mathcal{H}(a_2), ..., \mathcal{H}(a_n)\big)$ be obtained by thresholding at $0.5$; that is, $\forall i \in 1..n : \mathcal{H}(a_i) = \mathbb{I}(a_i > 0.5)$. Then we have $\mathcal{H}\big(\mathcal{R}_G(\boldsymbol{a})\big) = \mu_G\big(\mathcal{H}(\boldsymbol{a})\big)$ for all soft assignments $\boldsymbol{a} \in [0, 1]^n$.*

*Proof.* Proof by induction on the number of gates $N$ in $\mu_G$: for the base case (i.e. $N = 1$), the circuit $\mu_G$ simply consists of one gate. Depending on the type of this gate, we can have three possibilities:

(**And**) $\mathcal{H}\big(\min(a_1, ..., a_n)\big) = \min\big(\mathcal{H}(a_1), ..., \mathcal{H}(a_n)\big) = \textbf{And}\big(\mathcal{H}(a_1), ..., \mathcal{H}(a_n)\big)$

(**Or**) $\mathcal{H}\big(\max(a_1, ..., a_n)\big) = \max\big(\mathcal{H}(a_1), ..., \mathcal{H}(a_n)\big) = \textbf{Or}\big(\mathcal{H}(a_1), ..., \mathcal{H}(a_n)\big)$

(**Not**) $\mathcal{H}\big(\mathcal{N}(a)\big) = \mathcal{H}(1 - a) = 1 - \mathcal{H}(a) = \textbf{Not}\big(\mathcal{H}(a)\big)$

Now let us assume the lemma holds for any circuit with strictly less than $N$ gates. We want to prove it also holds for any circuit $\mu_G$ with $N$ gates. For the sake of simplicity, let us assume the sink node (i.e. the final gate) of $\mu_G$ is an **And** gate (the same argument can be made for **Or** and **Not** gates). If the final gate has $k$ inputs and is removed from the circuit, we will end up with $k$ (possibly overlapping) sub-circuits $\mu_{G_1}, ..., \mu_{G_k}$ with corresponding evaluator networks $\mathcal{R}_{G_1}, ..., \mathcal{R}_{G_k}$. We can then write:

$$\mathcal{H}\big(\mathcal{R}_G(\boldsymbol{a})\big) = \mathcal{H}\bigg(\min\big(\mathcal{R}_{G_1}(\boldsymbol{a}), ..., \mathcal{R}_{G_k}(\boldsymbol{a})\big)\bigg) = \textbf{And}\bigg(\mathcal{H}\big(\mathcal{R}_{G_1}(\boldsymbol{a})\big), ..., \mathcal{H}\big(\mathcal{R}_{G_k}(\boldsymbol{a})\big)\bigg)$$

$$= \textbf{And}\bigg(\mu_{G_1}\big(\mathcal{H}(\boldsymbol{a})\big), ..., \mu_{G_k}\big(\mathcal{H}(\boldsymbol{a})\big)\bigg) = \mu_G\big(\mathcal{H}(\boldsymbol{a})\big)$$

where the second and the third equalities come from the base and the inductive steps of the induction, respectively. □

**Theorem 2.** *$\mu_G$ is UNSAT if and only if $\mathcal{R}_G(\boldsymbol{a}) \leq 0.5$ for all soft assignments $\boldsymbol{a} \in [0, 1]^n$.*

*Proof.* (If) Proof by contradiction: let us assume $\mu_G$ is indeed SAT. Then there exists at least one hard assignment $\hat{\boldsymbol{a}} \in \{0, 1\}^n$ such that $\mu_G(\hat{\boldsymbol{a}}) = 1$. However, for hard assignment values, the $\min(\cdot)$, $\max(\cdot)$ and $\mathcal{N}(\cdot)$ functions behave exactly the same as the **And**, **Or** and **Not** gates, respectively. This means that for hard assignments, we have $\mu_G \equiv \mathcal{R}_G$, which further yields $\mathcal{R}_G(\hat{\boldsymbol{a}}) = \mu_G(\hat{\boldsymbol{a}}) = 1 > 0.5$. This would in turn lead to a contradiction.

(Only-If) Proof by contradiction: let us assume that there exists a soft assignment $\boldsymbol{a} \in [0, 1]^n$ such that $\mathcal{R}_G(\boldsymbol{a}) > 0.5$, then using Lemma 1 and the definition of $\mathcal{H}(\cdot)$, we will have:

$$\mu_G\big(\mathcal{H}(\boldsymbol{a})\big) = \mathcal{H}\big(\mathcal{R}_G(\boldsymbol{a})\big) = \mathbb{I}\big(\mathcal{R}_G(\boldsymbol{a}) > 0.5\big) = 1$$

In other words, we have found a hard assignment $\mathcal{H}(\boldsymbol{a})$ that satisfies the circuit $\mu_G$; this is a contradiction. $\qquad\square$

## APPENDIX C: CONVERTING CNF TO CIRCUIT

There are many ways one can convert a CNF to a circuit; some are optimized toward extracting structural information – e.g. Fu & Malik (2007). Here, we have taken a more intuitive and general approach based on the Cube and Conquer paradigm (Heule et al. (2011)) for solving CNF-SAT problems. In the Cube and Conquer paradigm, for a given input Boolean formula $F$, a variable $x$ in $F$ is picked and set to TRUE once to obtain $F_x^+$ and to FALSE the other time to get $F_x^-$. Now if we can find a SAT solution for either of $F_x^+$ or $F_x^-$, then we also have a SAT solution for $F$. Since neither of $F_x^+$ or $F_x^-$ contains $x$, we effectively reduce the complexity of the original SAT problem by removing one variable. This process can be repeated recursively (up to a fixed level) for $F_x^+$ and $F_x^-$ by picking a new variable to reduce the complexity even further. Now inspired by this paradigm, one can easily show that the following logical equivalence holds for any variable $x$ in $F$:

$$F \Leftrightarrow (x \wedge F_x^+) \vee (\neg x \wedge F_x^-) \tag{8}$$

And this is exactly the principle we used to convert a CNF formula $F$ into a circuit. In particular, by applying the equivalence in equation 8 recursively, up to a fixed level[4], we perform the CNF to circuit conversion (Note that $F_x^+$ and $F_x^-$ are also CNFs). The natural question then is in what order we should pick variables to apply equation 8. That is where the heuristic part comes into play: depending on the specific class of SAT problems we are targeting, we can incorporate a garden variety of ordering heuristics (aka *the branching heuristics*) in the literature – e.g. Biere et al. (2009); Heule et al. (2011); Marques-Silva (1999); Moskewicz et al. (2001). In our experiments for random $k$-SAT problems, each time we simply pick the variable that appears in the largest number of clauses in the current CNF.

## APPENDIX D: REPRESENTING GRAPH $k$-COLORING AS CNF AND CIRCUIT

We know from Computer Science theory that the graph $k$-coloring problem can be reduced to the SAT problem by representing the problem as a Boolean CNF. There are many ways in the literature to do so; we have picked the *Muldirect* approach from Velev (2007). In particular, for a graph with $N$ nodes and maximum $k$ allowed colors, we define the Boolean variables $x_{ij}$ for $1 \le i \le N$ and $1 \le j \le k$, where $x_{ij} = 1$ indicates that the $i$th node is colored by the $j$th color. Then, the CNF encoding the decision graph $k$-coloring problem is defined as:

$$\left[ \bigwedge_{i=1}^{N} \left( \bigvee_{j=1}^{k} x_{ij} \right) \right] \wedge \left[ \bigwedge_{(p,q) \in E} \left( \bigwedge_{j=1}^{k} (\neg x_{pj} \vee \neg x_{qj}) \right) \right] \tag{9}$$

where $E$ is the set of the graph edges. The left set of clauses in equation 9 ensure that each node of the graph takes *at least* one color. The right set of clauses in equation 9 enforce the constraint that the neighboring nodes cannot take the same color. As a result, any satisfiable solution to the CNF in equation 9 corresponds to at least one coloring solution for the original problem if not more. Note that in this formulation, we do not require each node to take only one color value; therefore, one SAT solution can produce multiple valid graph coloring solutions.

To generate a circuit from the above CNF, we note that the graph structure in graph coloring problem contains valuable structural information that can be potentially encoded as heuristics into the circuit structure. One such good heuristics, in particular, is the node degrees. More specifically, the *most constrained variable first heuristic* in Constraint Satisfaction Problems (CSPs) recommends assigning values to the most constrained variable first. In graph coloring problem, the higher the node degree, the more constrained the variables associated with that node are. Therefore, sorting the graph nodes based on their degrees would give us a meaningful variable ordering, which can be further used to build the circuit using the equivalence in equation 8, for example.

---

[4]Such that the whole process stays in the polynomial time range.

