# OpenReview forum: "Learning To Solve Circuit-SAT: An Unsupervised Differentiable Approach"
_ICLR.cc/2019/Conference_

### Official Review · AnonReviewer3 · 2018-11-01
**review for Circuit-SAT**

**Rating:** 7
**Confidence:** 3

**Review:**

The paper makes a nice contribution to solving Circuit-SAT problem from a Neuro-Symbolic approach, particularly, 1) a novel DAG embedding with a forward layer and a reverse layer that captures the structural information of a circuit-sat input. 2) Compared with Selsam et al.'s work of Neuro-SAT, the proposed model in this paper, DG-DAGRNN, directly produces an assignment of variables, and the method is unsupervised and end-to-end differentiable. 3) Empirical experiments on random k-SAT and random graph k-coloring instances that support the authors' claim on better generalization ability.

The paper is lucid and well written, I would support its acceptance at ICLR. Though I have a few comments and questions for the authors to consider.

- In figure 1 (a), what are x11, x12, etc?

- When comparing the two approaches of Neuro-Symbolic methods, besides the angles of optimality and training cost, it is worth to mention that the first one that based on classical algorithms always has a correctness guarantee, while the second one (learning the entire solution from scratch) usually does not.

- Section 4.1, as a pure decision problem, solving SAT means that giving a yes/no answer (i.e., a classification); while for practical purposes, solving SAT means that producing a model (i.e., a witness) of the formula if it is SAT. This can be misleading for some readers when the authors mentioning "solving SAT", and it would be clear if the authors could make a distinction when using such terms.

- Section 4.1, "without requiring to see the actual SAT solutions during training", again, what is the meaning of "solutions" is not very clear at this point. Readers may realize the experiments in the paper only train with satisfiable formulae from the afterward description, so the "solutions" indicates the assignments of variables. But it would be better to make it clear.

- Section 4.1/The Evaluator Network, "one can show also show that min() < S_min() <= S_max() < max()", what is the ordering relation (i.e., < and <=) here? It is a bit confusing if a forall quantifier for inputs (a_1, ... a_n) is required here.

- Section 4.1/The Evaluator Network, how does the temperature affect the results of R_G? It would be helpful to show their dynamics.

- Section 4.1/Optimization, "if the input circuit is UNSAT, one can show that the maximum achievable values for S_\theta is 0.5", it would be better to provide a brief description of how it is guaranteed. Also, this seems to be suggesting the DG-SAGRNN solver has no false positives, i.e., it will never produce a satisfiable result for unsatisfiable formulae? This would be interesting toward some semi-correctness if the answer is yes.

- Section 5.1, are the testing data all satisfiable formulae? If yes, then the figure 2 shows there is a number of satisfiable formulae but both the models cannot produce correct results -- is that a correct understanding of figure 2? If not, then what is the ground truth?

- I would love to see more experiments on SAT instances with a moderate number of variables but from real-world applications. It would be interesting to see how the model utilizes the rich structural information of instances from real applications (instead of randomly generated formulae).

- The training time and testing time(per instance) are not reported in the experiments.

---

> ### Author Response · Authors · 2018-11-10
> **Response to AnonReviewer3**
>
> - In figure 1 (a), what are x11, x12, etc?
> x represents the node feature vector that specify the nodes of the input DAG; in particular, x_{ij} is the j-th feature of the i-th node. As explained in the paper, in the Circuit-SAT problem, node feature vectors represent the type of node operation (i.e. AND, OR, NOT or VARIABLE) in the input circuit, represented as one-hot vectors.
>
> - Correctness guarantee:
> This is a great point indeed that we have briefly mentioned in the paper. In our framework, an input circuit is deemed as SAT if and only if the Solver network can produce an assignment that satisfies the Evaluator network. The test-time Evaluator network (or the train-time Evaluator network at low temperatures) mimics the exact behavior of the input circuit for continuous soft assignments; that is, if the soft assignment produced by the Solver network satisfies the test-time Evaluator network, its 0/1 hard counterpart will also satisfy the original circuit. Using this mechanism has two implications: (a) our framework does NOT produce false positives; if the input circuit is deemed as SAT, it means we have already found a satisfying solution for it. (b) if the satisfiability value for an input circuit is less than 0.5, all we can say about its SAT status is "unknown"; in other words, our method does not provide any proof of unsatisfiability.
>
> - Clarifying "SAT Solving":
> We have clarified this in the revised draft.
>
> - Clarifying "solutions":
> We have clarified this in the revised draft.
>
> - min() < S_min() < S_max() < max():
> Again the reviewer is correct and the ordering relation holds for all the inputs (a_1, ... a_n); we have clarified this in the revised draft.
>
> - The effect of temperature on the Evaluator network:
> In the beginning of the training, when the temperature is high, all the AND and OR gates in the Evaluator network (represented as S_min and S_max functions, respectively) act almost as arithmetic mean, so the training can be seen as maximizing the average values over the soft assignments (or their negations) while the gradient signal propagates back through all paths in the circuit; this is the exploration phase. As the training gradually progresses and the temperature anneals toward zero, the s_min and the s_max functions converge toward min and max functions, respectively, which in turn mimic the behavior of AND and OR gates for soft assignments. At this stage, the gradient signal will, for the most part, travel back only through the active paths in the circuit; this is the exploitation phase of learning.
>
> - Proving (circuit is UNSAT iff S_\theta <= 0.5 for all soft assignments):
> Here we give a sketch of the proof, but we are hoping to add an entire Appendix in the camera-ready version detailing the proof.
>
> (1) circuit is UNSAT if S_\theta <= 0.5 for all soft assignments:
> The proof can simply be achieved by contradiction.
>
> (2) S_\theta <= 0.5 for all soft assignments if circuit is UNSAT:
> The proof can be achieved by induction on the size (i.e. the number of gates) in the circuit: each time, isolate and remove the sink node of the circuit DAG (i.e. the last logical operator evaluated in the expression) and show that the output of the circuit is always less than or equal to 0.5 for all types of sink gates by assuming the statement of the theorem holds for the the resulted sub-circuits which have strictly smaller sizes compared to the original circuit.
>
> As for false positives, yes our model never produces false positives. Please refer to the above proof as well as the correctness explanation above.
>
> - Figure 2 explanation:
> Yes, all the test cases are SAT instances and, as the reviewer mentioned, there are some SAT examples where neither of the two models can decode within the allowed T_max iterations. We suspect these examples belong to the region of the K-SAT instances that's super close to SAT-UNSAT phase transition point. This region mostly contains the hardest K-SAT instances.
>
> - Real-world datasets:
> We completely agree with the reviewer that one of the main benefits of using learning methods for SAT solving is the ability of these methods to adapt to the target distribution of specific domains. This is indeed one of our current, ongoing efforts to adapt our framework to specific real-world domains. Nevertheless, we should emphasize that in our experiments, despite being random, both the training and the test examples are drawn from the hardest region of the SAT problems (the area close to SAT-UNSAT phase transition). This is achieved by using the data generation process proposed in the NeuroSAT paper.
>
> - The training time and testing time:
> Thank you for bringing up this important point. We have included a new paragraph in Section 5.1 detailing the time complexity.

---

### Official Review · AnonReviewer1 · 2018-11-02
**strong paper**

**Rating:** 8
**Confidence:** 4

**Review:**

The Authors of this paper investigate Neuro-Symbolic methods in the context of learning a SAT solver generalized to the Circuit-SAT problem. Using a reinforcement learning – inspired approach to demonstrate a framework that is capable of (unsupervised) learning, by means of an end-to-end differentiable training procedure. Their formulation incorporates the solving of a given SAT problem into the architecture, meaning the algorithm is trained to produce a solution if a given problem is satisfiable. This is in contrast to previous similar work by (Selsam et al. 2018), where the framework was trained as a SAT classifier. Their results outline the performance increase over the previous work (Selsam et al. 2018) on finding a given solution for a SAT problem, on in-sample and out-sample results.

Neg:
Figure descriptions are not very clear
When it comes to comparing the results, they do use a prepossessing step for their algorithm which they do not incorporate into the results

Pros:
Clear outline of the data sets used for benchmarks.
Good Literature review, expressing in-depth knowledge of the current state of the art formulation for same/similar tasks
Extensive background section, that explains the theoretical concepts and their architecture used well.
Clear outline of the Solver, where the individual parts/networks are explained and justified in detail
Very well outlined argumentation for approaching this particular problem by the proposed method/
The experimental results as well are easy to follow and show promising results for the proposed framework
The proposed method as well is novel and outperforms similar algorithms in the experimental evaluation.


The paper is very well written, proposes a novel Neuro-Symbolic  approach to the classical SAT problem, and demonstrates promising results.

---

> ### Author Response · Authors · 2018-11-10
> **Response to AnonReviewer1**
>
> We have been working to improve the clarity of the paper including the figures. So we are hoping the final version would address all the clarity concerns.
>
> As for pre-processing the input data, we only perform a CNF-to-Circuit conversion step as fully explained in Appendices B and C. As mentioned in the paper, although there are some problem domains where the input instances naturally come in the circuit format (e.g. circuit verification), in order to make a fair comparison with NeuroSAT, we decided to use the same CNF datasets that we used for NeuroSAT and as such we needed a pre-processing step to convert those CNFs to circuits. Nevertheless, we made sure this pre-processing step to be of O(N) in the worst case. As an extra advantage point, as mentioned in Appendix C, since our framework is capable of harnessing the circuit structure, domain-specific heuristics can be injected into the circuit structure during the pre-processing step - e.g. in graph k-coloring.

---

### Official Review · AnonReviewer2 · 2018-11-03
**Possibly interesting ideas, but needs more experiments**

**Rating:** 6
**Confidence:** 5

**Review:**

The paper proposes a graph neural network architecture that is designed to use the DAG structure in the input to learn to solve Circuit SAT problems. Unlike graph neural nets for undirected graphs, the proposed network propagates information according to the edge directions, using a deep sets representation to aggregate over predecessors of each vertex and GRUs to implement recurrent steps. The network is trained by using a "satisfiability function" which takes soft variable assignments computed by the network and applying a relaxed version of the circuit to be solved (replacing AND with softmax, OR with softmin, and NOT with 1 - variable value) to compute a continuous score that measures how satisfying the assignment is. Training is done by maximizing this score on a dataset of problem instances that are satisfiable. Results are shown on random k-SAT and graph coloring problems.

The paper is reasonably well-written and easy to follow. The idea of using the relaxed version of the circuit for training is nice. Combining ideas from DAG-RNNs and Deep Sets is interesting, although incremental.

Criticisms:
- How much does tailoring the network architecture to the DAG structure of the circuit actually help? A comparison to a regular undirected graph neural network on the circuit input without edge directions would be useful. In particular, since both edge directions are used in the current architecture but represented as two different DAGs, it naturally raises the question of whether a regular undirected graph neural net would also work well.
- How does the proposed approach compare to the current state-of-the-art non-learning approaches to SAT (CDCL, local search, etc.)? There is a huge literature on SAT, and ignoring all that work and comparing to only NeuroSAT seems unjustified. Without such comparisons, it is hard to say what is the benefit learning approaches in general, and the specific approach in this paper, provide in this domain. Even basic sanity-check baselines, e.g., random search, can be valuable given that the domain is somewhat new to learning approaches.
- One way to interpret the proposed approach is that it is learning to propose soft assignments that can be easily rounded. It would be good to compare to a Linear Programming relaxation-based approach that represents the SAT instance as an integer program with binary variables, relaxes the variables to be in [0,1], solves the resulting linear program, and rounds the solution. Do these approaches share the same failure modes, how does their performance differ, etc.
- The proposed approach has an obvious advantage over NeuroSAT in that it has access to the circuit structure, in addition to the flat representation of the SAT instance. According to the paper, not providing the circuit structure to the proposed approach hurts its performance. It would be useful to devise an experiment where a modified version of NeuroSAT is given the circuit structure as an additional input to see whether that closes the gap between the approaches.

---

> ### Author Response · Authors · 2018-11-10
> **Response to AnonReviewer2 (Part 1)**
>
> -Tailoring to DAG structure / directed vs undirected propagation:
> We would like to emphasize that our experimental setup does NOT aim at comparing the directed vs undiredted message passing on graphs. In particular, any form of message passing on graphs by definition imposes (momentary) directions on the edges of the graph even if the underlying graph is undirected; that is, message passing is always directed. On the other hand, what we are contrasting in this paper is *sequential* propagation based on some specific node order vs *synchronous* propagation based on no order. Furthermore, we argue the "specific order" for sequential propagation cannot be just any random order, but it has to arise from the semantics of the problem. In particular, in the Circuit-SAT problem, the node order (and its reverse version) is induced by the order by which the logical operators are evaluated in the circuit (i.e. the topological order of the input DAG). In theory, given unbounded training data and training time, one should still be able to learn the target Circuit-SAT function while ignoring this order and using synchronous propagation, but in practice with finite data and time, the learning is intractable for general circuits. In fact, before fully developing our DG-DAGRNN framework, we experimented with synchronous propagation for general circuits, but we were not able to learn the SAT function. The reason is somewhat intuitive: if we want to consume general (non-flat) circuits, ignoring the evaluation order of operators (i.e. using synchronous propagation) adds an extra task of figuring out the correct expression structure on the top of learning to solve the SAT problem itself which makes the learning task way more difficult. And that's why providing this structure explicitly via the DG-DAGRNN framework makes a huge improvement. In contrast, the synchronous propagation is NOT problematic for the CNF-SAT problem because the clauses in a flat CNF do not adhere any specific order and can be evaluated in any order, and therefore, synchronous propagation works well in NeuroSAT which only consumes CNFs.
>
> -Comparison against modern SAT solvers:
> This is indeed a very reasonable concern; nevertheless, we should emphasize that neither our framework nor NeuroSAT lay any claim to being on par with modern SAT-solvers at the moment. But that's not the goal here. This specific area in representation learning is relatively new and we are still in the feasibility study phase to see how much signal we can extract for SAT solving via deep learning. For practical purposes however, our intuition is that a successful approach that can potentially beat the classical solvers would be a hybrid of both learned models and traditional heuristic search components. But before getting there, we would need to gain a good understanding of what kind of useful signals we can or cannot extract from the problems structure via pure learning.
> That said, we have made a time comparison with MiniSAT (a popular, highly-optimized solver for moderate size problems). Even though, MiniSAT runs faster per example, our model, being a a neural network, is far more parallelizable and can solve many problems concurrently in a single batch. This would in turn make our method much faster than MiniSAT when applied on large sets of problems. We have included a new paragraph in the revised version describing this phenomenon.

---

> ### Author Response · Authors · 2018-11-10
> **Response to AnonReviewer2 (Part 2)**
>
> -SAT as an Integer Linear Program (ILP):
> Modeling the SAT problem as a (relaxed) ILP is a very interesting idea and there are some prior works on that in the literature. Nevertheless, such methodology would require solving an optimization problem for every problem instance at the test time. However, our proposed methodology is quite different (even though we also work with relaxed assignments): after training, our framework produces a recursive neural network (the Solver network) that can be run on test problem instances on GPU *without* needing to solve any optimization at the test time. That said, one interesting idea would be to replace our Evaluator network (i.e. the relaxed circuit) with a network that encodes the relaxed ILP and study the effects of that on training the Solver network. Exploring different options for the Evaluator network is indeed a future direction on our agenda.
>
> -A modified version of NeuroSAT to take in circuit structure:
> Our understanding is that the main ingredients that make NeuroSAT NeuroSAT are (a) a graph neural network for bi-partite graphs to embed the input CNF and (b) training this network toward SAT classification. In order for NeuroSAT to consume circuit structure, one would need to replace the first part with another sophisticated graph neural network that can process and understand variable-sized and topologically-diverse DAGs (circuits). But that's exactly what we have developed in this paper: the DG-DAGRNN architecture. So while we can in theory replace a fundamental ingredient of NeuroSAT with our proposed model, we are not sure we can still call the resulted framework NeuroSAT and close the gap. In other words, upgrading NeuroSAT to understand circuit structure is a non-trivial task and in fact one of the main contributions of the present work.

---

### Author Response · Authors · 2018-11-10
**Revision #1**

We would like to thank all the reviewers for bringing up some important questions and their detailed, constructive feedbacks. We have uploaded the first revised version of the paper addressing some of these concerns. In particular, the new draft includes:

1) A revised version of Figure 1 to fix an error in the figure.
2) A few clarifying statements to address some reviewers concerns regarding clarity.
3) A new paragraph detailing the time comparisons between the competing methods as well as the off-the-shelf MiniSAT solver.

In what follows, we will address the reviewers' questions and concerns in more details.

---

### Meta-Review · Area_Chair1 · 2018-12-13
**Good paper, comparison with traditional SAT solvers would be helpful**

**Confidence:** 4
**Recommendation:** Accept (Poster)

**Metareview:**

This paper introduces a new graph neural network architecture designed to learn to solve Circuit SAT problems, a fundamental problem in computer science. The key innovation is the ability to to use the DAG structure as an input, as opposed to typical undirected (factor graph style) representations of SAT problems. The reviewers appreciated the novelty of the approach as well as the empirical results provided that demonstrate the effectiveness of the approach.  Writing is clear. While the comparison with NeuroSAT is interesting and useful, there is no comparison with existing SAT solvers which are not based on learning methods. So it is not clear how big the gap with state-of-the-art is. Overall, I recommend acceptance, as the results are promising and this could inspire other researchers working on neural-symbolic approaches to search and optimization problems.